# Global prevalence and trends in hypertension and type 2 diabetes mellitus among slum residents: a systematic review and meta-analysis

Olalekan Abdulrahman Uthman ![ORCID],[1] Abimbola Ayorinde ![ORCID],[2] Oyinlola Oyebode,[1] Jo Sartori,[3] Paramjit Gill,[2] R J Lilford[3]

¹Warwick Centre for Global Health, Warwick Medical School, University of Warwick, Coventry, UK
²Division of Health Sciences, Warwick Medical School, University of Warwick, Coventry, UK
³Institute of Applied Health Research, University of Birmingham, Birmingham, UK

**Correspondence to**
Professor Olalekan Abdulrahman Uthman;
olalekan.uthman@warwick.ac.uk

## ABSTRACT

**Objective** First, to obtain regional estimates of prevalence of hypertension and type 2 diabetes in urban slums; and second, to compare these with those in urban and rural areas.

**Design** Systematic review and meta-analysis.

**Eligibility criteria** Studies that reported hypertension prevalence using the definition of blood pressure ≥140/90 mm Hg and/or prevalence of type 2 diabetes.

**Information sources** Ovid MEDLINE, Cochrane CENTRAL and EMBASE from inception to December 2020.

**Risk of bias** Two authors extracted relevant data and assessed risk of bias independently using the Strengthening the Reporting of Observational Studies in Epidemiology guideline.

**Synthesis of results** We used random-effects meta-analyses to pool prevalence estimates. We examined time trends in the prevalence estimates using meta-regression regression models with the prevalence estimates as the outcome variable and the calendar year of the publication as the predictor.

**Results** A total of 62 studies involving 108 110 participants met the inclusion criteria. Prevalence of hypertension and type 2 diabetes in slum populations ranged from 4.2% to 52.5% and 0.9% to 25.0%, respectively. In six studies presenting comparator data, all from the Indian subcontinent, slum residents were 35% more likely to be hypertensive than those living in comparator rural areas and 30% less likely to be hypertensive than those from comparator non-slum urban areas.

**Limitations of evidence** Of the included studies, only few studies from India compared the slum prevalence estimates with those living in non-slum urban and rural areas; this limits the generalisability of the finding.

**Interpretation** The burden of hypertension and type 2 diabetes varied widely between countries and regions and, to some degree, also within countries.

**PROSPERO registration number** CRD42017077381.

## INTRODUCTION

Non-communicable diseases (NCDs) are currently the leading cause of death globally; even in low/middle-income countries (LMICs), the burden of disease is shifting

### Strengths and limitations of this study

► To reduce the chance of missing relevant studies, no language constraints were applied during the literature search.
► The data were extracted by two independent reviewers, reducing the possibility of bias.
► We analysed trends over time, and between geographical regions.
► The substantial between-studies heterogeneity is an important limitation.
► Of the included studies, only few studies from India compared the slum prevalence estimates with those living in non-slum urban and rural areas; this limits the generalisability of the finding.

from infectious diseases to NCDs.[1] NCDs now account for about 41 million deaths annually, corresponding to nearly 7 in 10 of all deaths worldwide. Every year, 15 million people of ages 30–69 years die from these diseases, more than 85% of which are people living in LMICs. Most of the deaths from NCDs are caused by cardiovascular diseases, followed by cancer and respiratory diseases. NCDs affect people in all age groups, countries and geographical regions. The leading causes of these diseases include increased consumption of unhealthy foods, increased physical inactivity and population ageing.[2–4] These factors are mediated through metabolic risk factors for NCDs, the most common of which include hypertension and type 2 diabetes.[2–4]

Urbanisation is a global phenomenon that is occurring at a fast pace in most LMICs.[5 6] For more than 20 years, urban settlements have been increasing in population size because of fast growth in urban births, significant movement of people from rural areas and sustained integration of the global economy.[5 6] The United Nations defines slums as urban areas with overcrowding,

poor sanitation infrastructure, limited access to safe water, and/or poor structural quality of housing.[7 8] Slums are now an important component of today's urban settlements and likely continue to be for the foreseeable future.[7 8]

Despite increased global awareness about the presence and persistence of slums, and evidence that their populations are affected by different health problems and needs to other urban inhabitants, the health of their inhabitants is under-researched.[7–10] The health of the urban poor, people with low socioeconomic status living in urban areas, is usually conflated with that of slum residents. Although there is substantial overlap between these groups, there are also richer residents within slum neighbourhoods, as well as urban poverty occurring in non-slum urban areas. Health outcomes for these two groups may differ depending on whether deprivation is at the individual (urban poverty) or neighbourhood level (slum resident) due to neighbourhood effects.[7 8 11 12] For example, with respect to NCD risk factors, those residents in slums, whatever their personal socioeconomic status, may be more exposed to common physical environmental risk factors (for example: air pollution increasing risk of hypertension), social environmental risk factors (for example: crime rates which may increase stress and drive metabolic risk) or institutional risk factors (for example: stigma on the basis of their address reducing access to appropriate medical care). Many existing studies of NCD risk factors done in urban areas do not disaggregate the population's health data by slum and non-slum status to allow for the detection of intraurban health disparities that are due to neighbourhood effects rather than individual socioeconomic status.[13–22]

Understanding how the global challenges of hypertension, type 2 diabetes and rapid unplanned urbanisation intersect, by investigating whether the up to 1 billion people residing in slums[23] are succumbing to these important metabolic risk factors for NCD, will inform priorities for health services and health policy in LMICs. To fill this research gap, we therefore systematically gathered all the publications that relate to the burden of hypertension among slum residents to (1) assess the contemporary prevalence estimates of hypertension among slum residents; (2) compare the prevalence of hypertension and type 2 diabetes in slums with those in two other types of settlement, that is, non-slum urban and rural areas; and (3) assess the proportion of those with hypertension who were aware of their hypertensive status, those on treatment and those with blood pressure (BP) under control.

## METHODS

### Protocol and registration

The study background, rationale, and methods were specified in advance and documented in a protocol that was published in the PROSPERO register (CRD42017077381).

### Search and information sources

We searched Ovid MEDLINE, Cochrane CENTRAL and EMBASE from inception to December 2020 using the following keywords: slum, shanty town, ghetto, hypertension and type 2 diabetes. The search strategy for MEDLINE is shown in online supplemental annex 1.

### Eligibility criteria

We evaluated each identified study against the following predefined selection criteria:

► *Types of studies:* we included all studies (cross-sectional studies, retrospective or prospective cohort studies) that reported prevalence of hypertension and type 2 diabetes mellitus among slum residents as a primary or secondary outcome. No language, publication date or publication status restrictions were imposed.
► *Types of participants:* adult population (18 years and above) living in slums (as defined by the authors of the original studies included).
► *Types of interventions:* not applicable.
► *Types of outcomes:* essential hypertension (also called primary or idiopathic hypertension), defined as persistent (seated) systolic BP (SBP) of 140 mm Hg or greater or had diastolic BP (DBP) 90 mm Hg or greater regardless of age and sex. We excluded studies that included subjects with pregnancy-induced, pre-eclampsia, malignant, portal, pulmonary, renal, intracranial or ocular hypertension. We also excluded studies that used only self-reported measure, that is, deducible from the use of antihypertensive drugs or self-reported physician-diagnosed cases. If data were available, we noted (1) the percentage of those aware of their hypertension status, (2) on any antihypertensive treatment and (3) BP controlled to a target level. Awareness of hypertension was defined as self-reporting of any prior diagnosis of hypertension by a healthcare professional. Treatment of hypertension was defined as receiving prescribed antihypertensive medication for management of high BP at some time in the 1 year preceding the survey. Control of hypertension was defined as the proportion of patients reporting antihypertensive therapy with SBP of less than 140 mm Hg and DBP of less than 90 mm Hg.

Type 2 diabetes was defined based on measured fasting plasma glucose, or oral glucose tolerance test. Type 2 diabetes was diagnosed if the fasting blood glucose was ≥126 mg/dL (≥7.0 mmol/L) after an overnight fast for at least 8 hours, or random capillary blood glucose of ≥11.1 mmol/L or if the participant was taking treatment for type 2 diabetes.

### Study selection

Two reviewers (OAU, AA) independently evaluated the eligibility and methodological quality of the studies obtained from the literature searches. All articles yielded by the database search were initially screened by their titles and abstracts to obtain studies that met inclusion criteria. In cases of discrepancies, agreement was reached

by discussion with a third reviewer. Two reviewers (OAU, AA) independently evaluated the full-text articles of all identified citations to establish relevance of the article according to the prespecified criteria. In cases of discrepancies, agreement was reached by discussion with a third reviewer.

## Data collection process and data items

OAU extracted data, and AA and OO checked the extracted data. For each study that met the selection criteria, details extracted included year of publication, country of origin, study design, sample size, sampling strategy, study period, setting (rural/urban/slum), sociodemographic variables, prevalence estimates, etc.

## Risk of bias (quality) assessment

We used the Risk of Bias Assessment tool for Non-randomized Studies[24] to assess the risk of bias of included studies (see online supplemental box 1). The risk of bias in a study was graded as low, high or unclear on the basis of study features including the selection (selection of participants and confounding variables), performance (measurement of exposure), detection (blinding of outcome assessments), attrition (incomplete outcome data) and reporting (selective outcome reporting).

For each included study, we estimated the precision (C) or margin of error, considering the sample size (SS) and the observed prevalence (p) of hypertension among slum dwellers from the formula:

$$SS = Z^2 \times p \times (1 - p) / C^2 \qquad (1)$$

where Z was the z-value fixed at 1.96 across studies (corresponding to 95% CI). The desirable margin of error is 5% (0.05) or lower.

## Synthesis of results

For the meta-analysis, we used DerSimonian-Laird random-effects model[25] due to anticipated variations in study population, healthcare delivery systems and stage of epidemic transition to pool the hypertension and type 2 diabetes prevalence estimates. We performed leave-one-study-out sensitivity analysis to determine the stability of the results.[26] This analysis evaluated the influence of individual studies by estimating the pooled prevalence estimates in the absence of each study.[26] We assessed heterogeneity among studies by inspecting the forest plots and using the $X^2$ test for heterogeneity with a 10% level of statistical significance and using the $I^2$ statistic where we interpret a value of 50% as representing moderate heterogeneity.[27 28] We assessed the possibility of publication bias by evaluating a funnel plot for asymmetry. Because graphical evaluation can be subjective, we also conducted an Egger's regression asymmetry test as formal statistical tests for publication bias.[29]

Following the overall analyses, we performed the following subgroup analyses: place of residence (rural vs urban slum vs non-slum urban); participants' risk factors, including socioeconomic position; study design

(cross-sectional, cohort); study location (low/middle-income vs high-income countries) and study precision.

We examined time trends in the prevalence estimates using meta-regression regression models with the prevalence estimates as the outcome variable and the calendar year of the publication as the predictor. In order to measure secular patterns in prevalence figures, we use the annual average percentage change (AAPC). We fitted a regression line to the natural logarithm of the prevalence estimates, that is, y=α+βx+ε, where y=ln(Prevalence), and x=calendar year. The AAPC was calculated as 100×(exp(β)−1). The 95% CI of the AAPC was also computed from the regression model.[30] The prevalence calculations indicated an upward trend when both the AAPC estimate and the lower limit of its 95% CI were >0. However, they indicated a downward trend when both the AAPC and its upper limits were less than 0. The prevalence estimates were otherwise considered stable over time.[30] This systematic review was reported according to the Preferred Reporting Items for Systematic Reviews and Meta-analyses guideline (online supplemental annex 2).[31]

## Patient and public involvement

No patient was involved.

# RESULTS
## Study selection and characteristics

The literature search yielded 1490 articles. Online supplemental figure 1 shows the study selection flow diagram. After review, 135 articles were selected for critical reading. Seventy-two studies did not meet the inclusion criteria and were excluded (see online supplemental table 1 for list of excluded studies). The other 62 studies involving 108 110 participants met the inclusion criteria and were included in the meta-analysis.[13–22 32–80] Forty-three studies reported only hypertension prevalence estimates, 29 studies reported only type 2 diabetes prevalence estimates and 8 reported both. Table 1 and online supplemental table 2 present the characteristics of the included studies. The studies were reported between 1989 and 2019. Studies were reported as full-text journal articles (n=61, 98%); except for one which was reported as a conference abstract. The number of participants included in the studies ranged from 100 to 15 763. When reported, the mean age of participants ranged from 32 years to 47 years. Most of the studies were carried out in South Asia: India (n=30); Bangladesh (n=8), Nepal (n=1) and Pakistan (n=1); followed by sub-Saharan Africa: Kenya (n=9) and Nigeria (n=4); Latin America and Caribbean: Brazil (n=5) and Peru (n=1); and East Asia and Pacific: Thailand (n=1). Most of the studies were conducted in the following urban slums: Kibera (n=4), Delhi (n=3), Hyderabad (n=3), Ajegunle (n=2), Chandigarh (n=2), Chennai (n=2), Dhaka (n=2), Haryana (n=2) and Maceio (n=2).

**Table 1** Pooled prevalence by different subgroups

| Subgroup | | Hypertension | | | Type 2 diabetes | | |
|---|---|---|---|---|---|---|---|
| | | n | % | I² | n | % | I² |
| Sample size | Smaller studies (<1000) | 27 | 25.9 (21.6 to 30.6) | 97.1 | 15 | 11.0 (8.2 to 14.2) | 93.9 |
| Sample size | Larger studies (1000+) | 17 | 21.4 (17.2 to 26.1) | 99.6 | 15 | 7.8 (5.1 to 11.1) | 99.4 |
| Study precision | Imprecise studies | 8 | 33.4 (25.7 to 41.7) | 91.2 | 1 | 25.2 (17.3 to 34.2) | – |
| Study precision | Precise studies | 36 | 22.3 (18.9 to 25.9) | 99.2 | 29 | 8.9 (6.9 to 11.2) | 98.9 |
| Publication year | 2001–2005 | 5 | 15.6 (9.0 to 23.8) | 94.7 | 4 | 8.2 (6.7 to 9.8) | 53.6 |
| Publication year | 2006 –2010 | 6 | 28.6 (18.9 to 39.4) | 98.7 | 4 | 6.3 (3.3 to 10.3) | 90.6 |
| Publication year | 2011–2020 | 33 | 24.7 (21.0 to 28.6) | 99.2 | 22 | 10.2 (7.4 to 13.4) | 99.2 |
| Region | South Asia | 27 | 25.1 (20.7 to 29.8) | 98.9 | 19 | 11.9 (9.1 to 15.1) | 97.6 |
| Region | Sub-Saharan Africa | 10 | 24.4 (17.7 to 31.9) | 99.2 | 8 | 4.5 (2.4 to 7.2) | 98.8 |
| Region | Latin America and Caribbean | 6 | 18.3 (13.4 to 23.9) | 97.1 | 1 | 10.2 (8.1 to 12.3) | – |
| Region | Middle East and North Africa | 1 | 31.2 (28.4 to 34.1) | – | 1 | 8.8 (7.1 to 10.6) | – |
| Region | East Asia and Pacific | – | – | – | 1 | 7.9 (6.3 to 9.7) | |
| Income category | Lower middle income | 36 | 25.2 (21.2 to 29.4) | 99.1 | 28 | 9.3 (7.0 to 11.92) | 98.9 |
| Income category | Upper middle income | 5 | 17.9 (12.1 to 24.6) | 97.6 | 2 | 9.0 (6.9 to 11.3) | 62 |
| Income category | Low income | 2 | 24.0 (16.9 to 32.0) | 92.2 | | | |
| Sex | Male | 24 | 22.5 (16.0 to 29.7) | 99.2 | 11 | 8.1 (5.1 to 11.6) | 97.6 |
| Sex | Female | 24 | 23.2 (18.6 to 28.1) | 98.7 | 11 | 7.3 (4.6 to 10.6) | 97.5 |
| Age | Young adult | 8 | 15.7 (10.1 to 22.1) | 97.8 | 2 | 2.1 (0.3 to 5.4) | 96.7 |
| Age | Middle-aged adult | 9 | 35.0 (25.0 to 45.6) | 99.2 | 2 | 5.6 (4.5 to 6.8) | 0 |
| Age | Older adult | 9 | 49.6 (36.7 to 62.6) | 98.3 | 2 | 9.1 (7.0 to 11.4) | 0 |
| Body mass index | Underweight | 5 | 21.8 (11.4 to 34.4) | 87.3 | | | |
| Body mass index | Normal weight | 6 | 21.9 (11.8 to 34.2) | 98.6 | 2 | 2.3 (1.8 to 2.8) | 0 |
| Body mass index | Overweight | 6 | 32.9 (21.2 to 45.8) | 97.4 | 2 | 4.2 (1.2 to 8.8) | 50 |
| Body mass index | Obese | 6 | 45.4 (34.5 to 56.6) | 93.3 | 2 | 6.4 (4.0 to 9.3) | 0 |
| Education status | Never studied | 7 | 39.1 (27.5 to 51.3) | 98 | 1 | 5.1 (3.0 to 7.8) | – |
| Education status | Less than primary | 4 | 18.3 (13.9 to 23.1) | 87.1 | 1 | 4.6 (3.4 to 6.1) | – |
| Education status | Primary | 6 | 24.8 (12.0 to 40.4) | 99.4 | 1 | 4.4 (3.6 to 5.2) | – |
| Education status | Secondary or higher | 7 | 22.4 (11.1 to 36.2) | 99.3 | 1 | 4.1 (3.2 to 5.2) | – |
| Income | Poorest | 5 | 20.9 (10.4 to 33.8) | 98.9 | | | |
| Income | Middle | 5 | 25.3 (10.6 to 43.8) | 99.5 | | | |
| Income | Least poor | 5 | 29.2 (13.1 to 48.5) | 98.3 | | | |
| Smoking status | Yes | 5 | 38.0 (19.1 to 59.0) | 99.1 | | | |
| Smoking status | No | 5 | 30.5 (17.6 to 45.2) | 99.6 | | | |
| Alcohol consumption | Yes | 3 | 26.5 (18.0 to 35.9) | 83.4 | | | |
| Alcohol consumption | No | 3 | 29.1 (9.3 to 54.3) | 99.7 | | | |
| Physically active | Yes | 3 | 28.8 (11.1 to 50.8) | 99.6 | | | |
| Physically active | No | 3 | 30.8 (7.7 to 60.9) | 98.4 | | | |
| Treatment cascade | Aware of HBP | 12 | 33.6 (19.1 to 50.0) | 99.7 | | | |
| Treatment cascade | On treatment | 9 | 51.9 (35.2 to 68.3) | 98.6 | | | |
| Treatment cascade | BP controlled | 8 | 25.9 (18.4 to 34.3) | 87.8 | | | |

World Bank Country Income Groups, 2018.
Participants were divided into age groups that, broadly defined, covered young adulthood (18–35 years), middle age (36–55 years) and older adulthood (56 years and older).
Underweight—body mass index under 18.5 kg/m².
Normal weight—body mass index greater than or equal to 18.5–24.9 kg/m².
Overweight—body mass index greater than or equal to 25–29.9 kg/m².
Obesity—body mass index greater than or equal to 30 kg/m².
Physical activity as defined by authors.
Alcohol consumption as defined by authors.
Smoking status as defined by authors.
Income status as reported by authors.
BP, blood pressure; HBP, high BP.

## Risk of bias of included studies

Summary of risk of bias assessment for each study is shown in online supplemental table 3. The risk of bias in the selection of participants was low in most studies (n=56, 90%), high in three studies (5%) and unclear in three studies (5%). Risk of bias due to confounding variables was low in most studies (n=39, 63%), high in 22 studies (36%) and unclear in 1 study. Risk of bias due to measurement of exposure, blinding of outcome assessments and selective outcome reporting was low in all the 62 studies as we included all studies that used objective measure of hypertension and type 2 diabetes. Risk of bias due to incomplete outcome data was low in most studies (n=54, 87%), high in two studies (3%) and unclear in six studies (10%).

## Variations in prevalence of hypertension and type 2 diabetes by geographical regions

Prevalence of hypertension and type 2 diabetes from individuals is shown in figures 1 and 2, respectively.

### East Asia and Pacific

*Thailand:* one study from Klong-Toey slum found that 77 of the 976 respondents had type 2 diabetes in 1989 (7.9%, 95% CI 6.3% to 9.8%).

### Latin America and Caribbean

*Brazil:* four studies reported the prevalence of hypertension from three different slums: Maceio (n=2), Rio de Janeiro (n=1) and Salvador (n=1). Florencio et al[42] found that almost one-third of the Maceio slum dwellers were hypertensive in 2004 (29.8%, 95% CI 24.8% to 35.2%), while Ferriera et al[41] estimated prevalence of hypertension among Maceio slum residents to be 14.8% (95% CI 10.4% to 20.2%) in 2005. The reported prevalence of hypertension in other slums was 11.3% (95% CI 10.2% to 12.4%) in Rio de Janeiro in 2007 and 20.6% (95% CI 19.5% to 21.7%) in Salvador in 2015. The pooled prevalence ('annualised year average') of hypertension for the four studies yielded an estimate of 18.4% (95% CI 12.0% to 26.2%). One study from Brazil found that 1 in 10 had type 2 diabetes in 2017.

*Peru:* one study from a Lima slum conducted in 2014 found that 21 of the 142 respondents were hypertensive (14.8%, 95% CI 9.4% to 21.7%).

### South Asia

*Bangladesh:* four studies from Dhakan slums reported prevalence of hypertension. The reported prevalence of hypertension ranged from 11.6% (95% CI 9.7% to 13.8%) in 2012 to 19.56% (95% CI 17.85% to 21.37%) in 2018. Five studies from Dhakan slums reported prevalence of type 2 diabetes. The pooled prevalence ('annualised year average') of hypertension for the three studies yielded an estimate of 16.1% (95% CI 12.2% to 20.3%). The reported prevalence of type 2 diabetes in these slums ranged from 8.1% (95% CI 6.8% to 9.6%) in 2004 to 18.12% (95% CI 16.46% to 19.87%) in 2019.

| Study | HTN | Total | Events per 100 observations | Prevalence | (95% CI) |
|---|---|---|---|---|---|
| **India** | | | | | |
| Lubree 2002 | 6 | 142 | | 4.23 | [1.57; 8.97] |
| Uthakalla 2012 | 30 | 400 | | 7.50 | [5.12; 10.53] |
| Misra 2001 | 62 | 532 | | 11.65 | [9.05; 14.69] |
| Singh 2012 | 510 | 3118 | | 16.36 | [15.07; 17.70] |
| Anand 2007 | 422 | 2562 | | 16.47 | [15.05; 17.97] |
| Chakerborty 2012 | 95 | 470 | | 20.21 | [16.67; 24.13] |
| Vikram 2003 | 136 | 639 | | 21.28 | [18.17; 24.66] |
| Vigneswari 2014 | 128 | 529 | | 24.20 | [20.61; 28.08] |
| Joshi 2013 | 24 | 100 | | 24.00 | [16.02; 33.57] |
| Dwivedi 2018 | 107 | 423 | | 25.30 | [21.22; 29.72] |
| Nirmala 2014 | 185 | 700 | | 26.43 | [23.20; 29.86] |
| Ahmad 2014 | 54 | 196 | | 27.55 | [21.42; 34.37] |
| Deepa 2011 | 4839 | 15763 | | 30.70 | [29.98; 31.43] |
| Chaturvedi 2007 | 188 | 596 | | 31.54 | [27.83; 35.44] |
| Acharyya 2014 | 360 | 1052 | | 34.22 | [31.35; 37.18] |
| Kar 2010 | 53 | 150 | | 35.33 | [27.71; 43.55] |
| George 2019 | 1311 | 3693 | | 35.50 | [33.95; 37.07] |
| Kar 2008 | 148 | 382 | | 38.74 | [33.83; 43.83] |
| Banerjee 2016 | 4304 | 10167 | | 42.33 | [41.37; 43.30] |
| Kumari 2014 | 76 | 174 | | 43.68 | [36.19; 51.39] |
| Sinha 2010 | 123 | 275 | | 44.73 | [38.75; 50.82] |
| Gonmei 2018 | 100 | 202 | | 49.50 | [42.41; 56.61] |
| Random effects model | | . | | 26.76 | [21.53; 32.33] |
| **Nigeria** | | | | | |
| Akinwale 2013 | 312 | 2434 | | 12.82 | [11.52; 14.21] |
| Sowemimo 2015 | 267 | 806 | | 33.13 | [29.88; 36.50] |
| Daniel 2013 | 368 | 964 | | 38.17 | [35.10; 41.33] |
| Ezeala–Adikaibe 2016 | 406 | 774 | | 52.45 | [48.87; 56.02] |
| Random effects model | | . | | 33.14 | [17.52; 50.96] |
| **Peru** | | | | | |
| Heitzinger 2014 | 21 | 142 | | 14.79 | [9.39; 21.71] |
| Random effects model | | . | | 14.79 | [9.38; 21.14] |
| **Nepal** | | | | | |
| Oli 2013 | 193 | 689 | | 28.01 | [24.69; 31.53] |
| Random effects model | | . | | 28.01 | [24.72; 31.43] |
| **Brazil** | | | | | |
| Marins 2007 | 369 | 3279 | | 11.25 | [10.19; 12.39] |
| Ferreira 2005 | 33 | 223 | | 14.80 | [10.41; 20.15] |
| Unger 2015 | 1162 | 5649 | | 20.57 | [19.52; 21.65] |
| Florencio 2004 | 94 | 315 | | 29.84 | [24.84; 35.23] |
| Random effects model | | . | | 18.55 | [11.45; 26.91] |
| **Kenya** | | | | | |
| van de Vijver 2013 | 640 | 5190 | | 12.33 | [11.45; 13.26] |
| Joshi 2014 | 258 | 2045 | | 12.62 | [11.21; 14.13] |
| Ongeti 2013 | 52 | 400 | | 13.00 | [9.86; 16.70] |
| Vusirikala 2019 | 751 | 3063 | | 24.52 | [23.00; 26.08] |
| Olack 2015 | 418 | 1528 | | 27.36 | [25.13; 29.67] |
| Edwards 2015 | 613 | 2206 | | 27.79 | [25.93; 29.71] |
| Random effects model | | . | | 19.16 | [13.38; 25.69] |
| **Bangladesh** | | | | | |
| Huda 2012 | 116 | 1000 | | 11.60 | [9.68; 13.75] |
| Rawal 2017 | 69 | 505 | | 13.66 | [10.79; 16.97] |
| Choudhury 2018 | 393 | 2009 | | 19.56 | [17.85; 21.37] |
| Khalequzzaman 2017 | 500 | 2551 | | 19.60 | [18.08; 21.19] |
| Random effects model | | . | | 16.06 | [12.20; 20.35] |
| **Egypt** | | | | | |
| Gadallah 2018 | 307 | 984 | | 31.20 | [28.31; 34.20] |
| Random effects model | | . | | 31.20 | [28.34; 34.13] |
| **Haiti** | | | | | |
| Tymejczyk 2019 | 181 | 894 | | 20.25 | [17.66; 23.03] |
| Random effects model | | . | | 20.25 | [17.67; 22.95] |

Test for subgroup differences: $\chi^2_8 = 64.13$, df = 8 (p < 0.01)

**Figure 1** Hypertension (HTN) prevalence estimates among slum residents and 95% CIs from individual studies and pooled data.

*India:* 22 studies from India reported prevalence of hypertension from more than 15 different slums. The reported prevalence varied across and within the slums. For example, Kar et al[48] estimated the prevalence of hypertension to be 27.6% (95% CI 21.4% to 34.4%) among 196 Chandigarh and Haryana slum residents in 2008; however, they estimated the prevalence of hypertension to be 16.5% (95% CI 15.1% to 18.0%) among 2 562 196 Chandigarh and Haryana slum residents in 2010. Prevalence of type 2 diabetes also varied across slums in India. The pooled prevalence ('annualised year average')

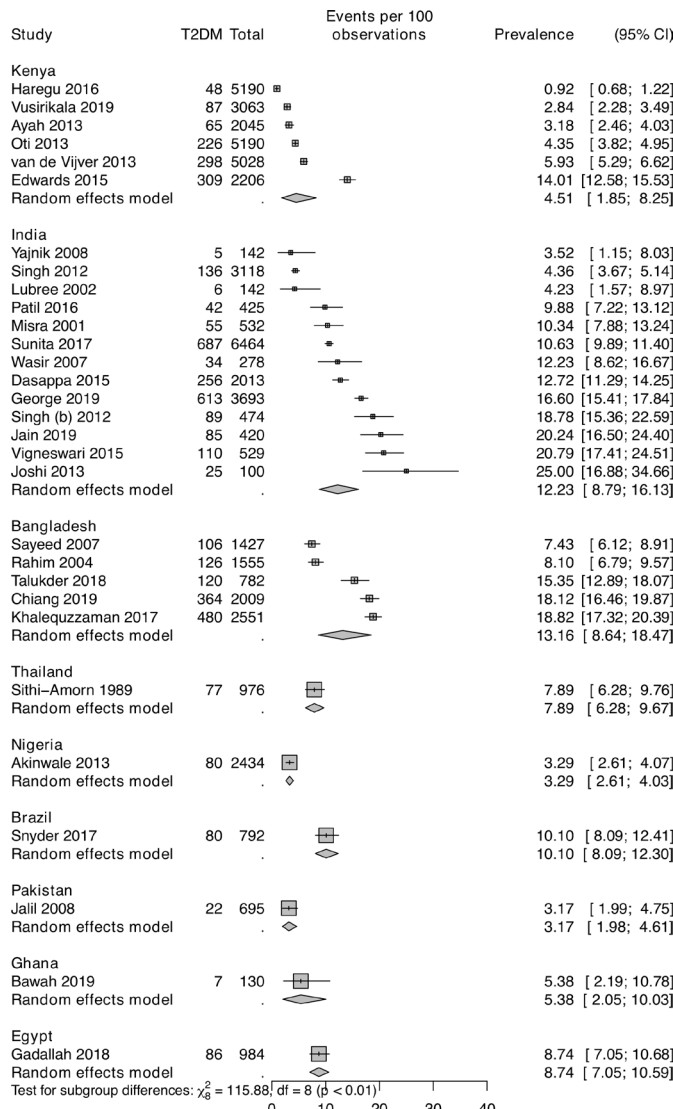

**Figure 2** Type 2 diabetes mellitus (T2DM) prevalence estimates among slum residents and 95% CIs from individual studies and pooled data.

of hypertension for the 22 studies yielded an estimate of 26.8% (95% CI 22.5% to 31.3%). In Delhi, the reported prevalence of type 2 diabetes ranged from 12.7% (95% CI 11.3% to 14.2%) in 2007 to 31.5% (95% CI 27.8% to 35.4%) in 2012. The pooled prevalence ('annualised year average') of type 2 diabetes for the 13 studies yielded an estimate of 12.2% (95% CI 9.2% to 15.6%).

*Nepal:* one study from a Kathmandu slum conducted in 2013 found that 193 of the 689 respondents were hypertensive (28.0%, 95% CI 24.7% to 31.5%).

*Pakistan:* one study from a Lahore slum found that 22 of the 695 respondents had type 2 diabetes in 2008 (3.2%, 95% CI 2.0% to 4.8%).

Sub-Saharan Africa, *Kenya:* six studies reported the prevalence of hypertension from three different slums: Kibera (n=4) and Viwandani and Korogocho (n=2). The reported prevalence among Kibera slum residents ranged from 13.0% (95% CI 9.9% to 16.7%) in 2013 to 27.8% (95% CI 25.9% to 29.7%) in 2015. van de Vijver *et al*[68]

found that 640 of the 5190 respondents from Viwandani and Korogocho slums were hypertensive (12.3%, 95% CI 11.5% to 13.3%). The pooled prevalence ('annualised year average') of hypertension for the six studies yielded an estimate of 19.2% (95% CI 13.2% to 26.0%). The reported prevalence of type 2 diabetes ranged from 0.9% (95% CI 0.7% to 1.2%) in Nairobi slum in 2016 to 4.4% (95% CI 3.8% to 5.0%) in Viwandani and Korogocho in 2013. The pooled prevalence ('annualised year average') of type 2 diabetes for the six studies yielded an estimate of 4.5% (95% CI 2.0% to 7.9%).

*Nigeria:* four studies from five different slums reported prevalence of hypertension. The reported prevalence varied across and within the slums. Ezeala-Adikaibe *et al*[40] found that half of the respondents from Enugu slums were hypertensive in 2016 (52.5%, 95% CI 48.9% to 56.0%). While Daniel *et al* and Sowemimo *et al*[16 64] found that almost one-third of the Ajegule (38.2%, 95% CI 35.1% to 41.3%, 2013) and Yemetu (33.1%, 95% CI 30.0% to 36.5%, 2015) slum residents were hypertensive. However, Akinwale *et al*[33] found that only 12.8% of the respondents from Ijora Oloye, Ajegunle and Makoko were hypertensive in 2013. The pooled prevalence ('annualised year average') of hypertension for the four studies yielded an estimate of 33.2% (95% CI 15.6% to 53.5%). Akinwale *et al* found that only 3.3% of the respondents from Ijora Oloye, Ajegunle and Makoko had type 2 diabetes in 2013.

### Secular trends in hypertension and type 2 diabetes prevalence estimates

Secular trends in hypertension, in five countries for which there were data across multiple time points, and type 2 diabetes, in three countries in which we had data across multiple time points, among slum residents are shown in figures 3 and 4. We observed a continuous increase in prevalence of hypertension among slum residents in four out of five countries. The increase is more pronounced in India, followed by Kenya and Bangladesh. The prevalence of hypertension increased by 204.6% from 11.7% in 2001 to 35.5% in 2019 in India. The prevalence of hypertension increased by 98.8% from 12.3% in 2013 to 24.5% in 2019 in Kenya. However, the results of the trend analysis showed statistically significant upward trends only in India, such that the prevalence of hypertension increased +6.9% (95% CI +2.0% to +12.0%) per year between 2001 and 2019. There was no statistically significant trend observed in Brazil using trend analyses (trend=−0.0%, 95% CI −22.7% to +29.2%). We also observed a continuous increase in prevalence of type 2 diabetes among slum residents in India and Bangladesh. The prevalence of type 2 diabetes increased by 123.6% from 8.1% in 2004 to 18.1% in 2019 in Bangladesh. The prevalence of type 2 diabetes increased by 95.8% from 10.3% in 2001 to 20.2% in 2019 in India. However, the results of the trend analysis showed statistically significant upward trends only in Bangladesh such that the prevalence of type 2 diabetes increased +5.9% (95% CI +1.1% to +10.8%) per year between 2004 and 2019. A non-statistically significant downward trend

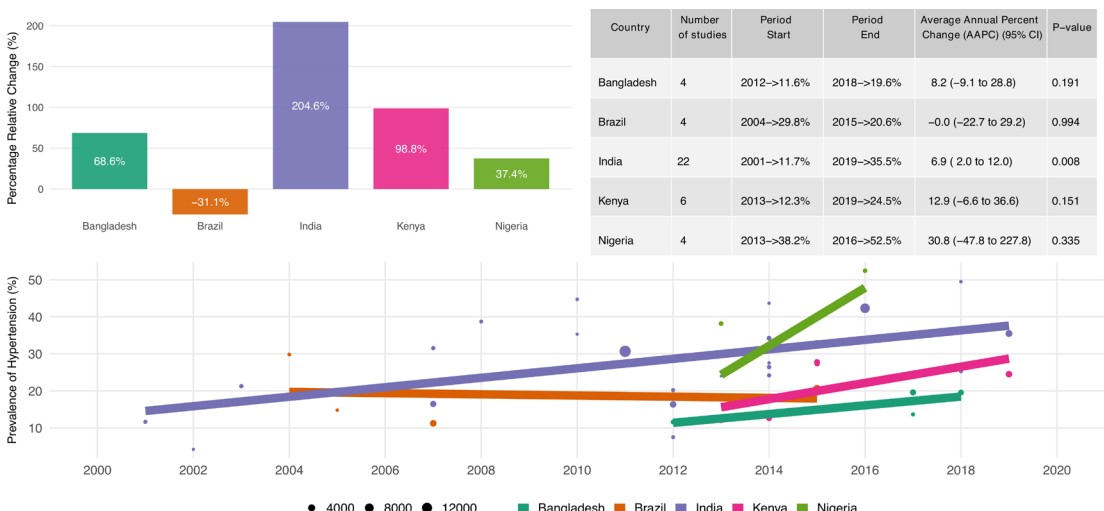

| Country | Number of studies | Period Start | Period End | Average Annual Percent Change (AAPC) (95% CI) | P–value |
|---|---|---|---|---|---|
| Bangladesh | 4 | 2012→11.6% | 2018→19.6% | 8.2 (−9.1 to 28.8) | 0.191 |
| Brazil | 4 | 2004→29.8% | 2015→20.6% | −0.0 (−22.7 to 29.2) | 0.994 |
| India | 22 | 2001→11.7% | 2019→35.5% | 6.9 ( 2.0 to 12.0) | 0.008 |
| Kenya | 6 | 2013→12.3% | 2019→24.5% | 12.9 (−6.6 to 36.6) | 0.151 |
| Nigeria | 4 | 2013→38.2% | 2016→52.5% | 30.8 (−47.8 to 227.8) | 0.335 |

**Figure 3**  Secular trends in hypertension prevalence estimates among slum residents across different regions.

in type 2 diabetes prevalence was also observed in Kenya (trend=−11.1%, 95% CI −45.7% to +45.6%).

## Prevalence of hypertension by different hypertension and type 2 diabetes subgroups

### Study characteristics

As shown in table 1, the pooled prevalence of hypertension was higher in studies conducted in lower middle-income countries (23.2%, 95% CI 21.5% to 29.0%, 36 studies) than those from upper middle-income countries (17.9%, 95% CI 12.1% to 24.6%, 5 studies). The pooled prevalence of hypertension tended to be higher among studies from South Asia (25.3%, 95% CI 21.3% to 29.6%, 26 studies) and sub-Saharan Africa (24.4%, 95% CI 17.7% to 31.9%, 10 studies) than those from Latin America and Caribbean (18.3%, 95% CI 13.4% to 23.9%, 6 studies). The pooled prevalence tended to be higher among imprecise studies (33.4%, 95% CI 25.7% to 41.7%, 8 studies) than those from precise studies (22.4%, 95% CI 18.9% to 26.1%, 35 studies). The pattern was similar for type 2 diabetes prevalence estimates.

### Sociodemographic characteristics

As shown in table 1, the pooled prevalence of hypertension was similar among men (22.5%, 95% CI 16.0% to 29.7%, 24 studies) and women (23.5%, 95% CI 18.6% to 28.1%, 24 studies). The pooled prevalence of hypertension tended to be higher among older adults (49.6%, 95% CI 36.7% to 62.6%, 9 studies) than middle-aged (35.0%, 95% CI 25.0% to 45.6%, 9 studies) and young adults (15.7%, 95% CI 10.1% to 22.1%, 8 studies). Similarly, the pooled prevalence of hypertension tended to be higher in obese (45.4%, 95% CI 34.5% to 56.5%, 6 studies) and overweight (32.9%, 95% CI 21.2% to 45.8%, 6 studies) participants than participants with normal (21.9%, 95% CI 11.8% to 34.2%, 6 studies) and underweight (21.8%, 95% CI 11.4% to 34.4%, 5 studies). The pooled prevalence of hypertension tended to be higher among those who never studied (39.1%, 95% CI 27.5% to 51.3%) than those with less than primary (18.3%, 95% CI 13.9% to 23.1%, 4 studies), primary (24.8%, 95% CI 12.0% to 40.4%, 6 studies) or secondary/higher

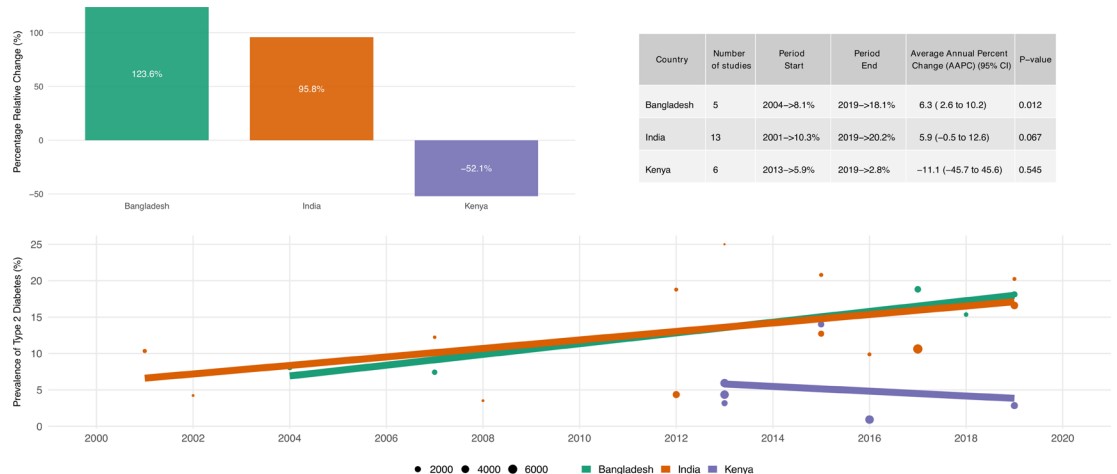

| Country | Number of studies | Period Start | Period End | Average Annual Percent Change (AAPC) (95% CI) | P–value |
|---|---|---|---|---|---|
| Bangladesh | 5 | 2004→8.1% | 2019→18.1% | 6.3 ( 2.6 to 10.2) | 0.012 |
| India | 13 | 2001→10.3% | 2019→20.2% | 5.9 (−0.5 to 12.6) | 0.067 |
| Kenya | 6 | 2013→5.9% | 2019→2.8% | −11.1 (−45.7 to 45.6) | 0.545 |

**Figure 4**  Secular trends in type 2 diabetes mellitus prevalence estimates among slum residents across different regions.

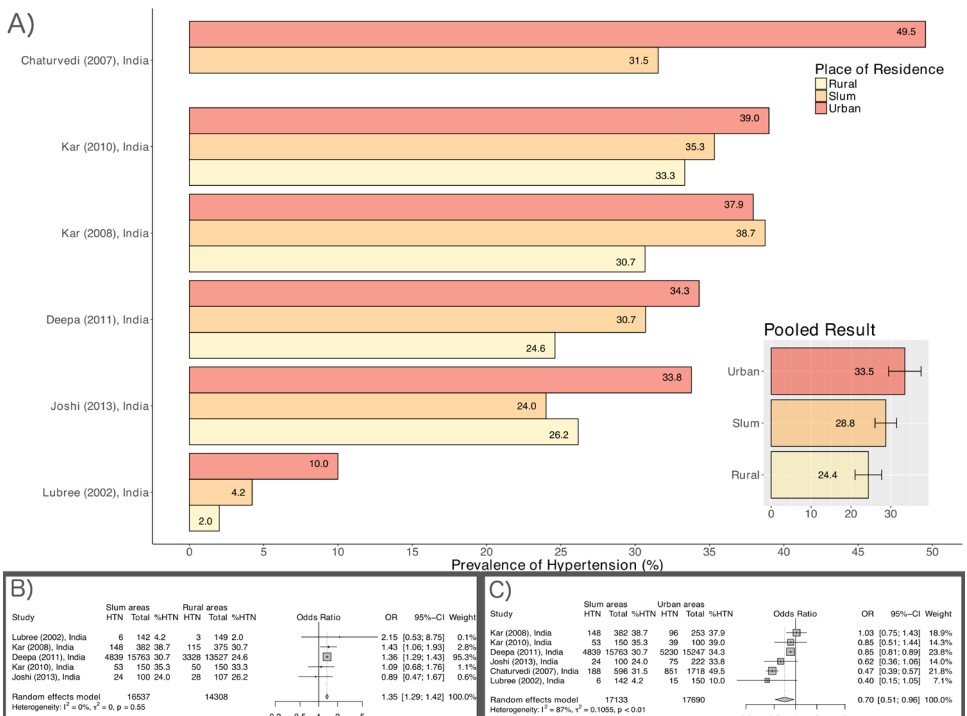

**Figure 5** Hypertension (HTN) prevalence estimates by place of residence: urban versus rural versus slum. (A) Data from each studies, (B) Pooled estimates by place of residence, (C) Comparative pooled estimates.

educational attainment (22.4%, 95% CI 11.2% to 36.2%, 7 studies). The pooled prevalence of hypertension tended to be higher among the least poor (29.2%, 95% CI 13.1% to 48.5%, 5 studies) than those with middle (25.3%, 95% CI 10.6% to 43.8%, 5 studies) and poorest income (20.9%, 95% CI 10.4% to 33.8%, 5 studies). The pattern was similar for type 2 diabetes prevalence estimates.

### Lifestyle factors

The pooled prevalence of hypertension tended to be higher among smokers (38.0%, 95% CI 19.1% to 59.0%, 5 studies) than those not smoking (30.5%, 95% CI 17.6% to 45.2%, 5 studies). We found that the pooled prevalence of hypertension tended to be higher for those not physically active (30.8%, 95% CI 7.7% to 60.9%, 3 studies) than those physically active (28.8%, 95% CI 11.1% to 50.8%); tended to be higher among those with no history of alcohol consumption (29.1%, 95% CI 9.3% to 54.3%, 3 studies) than those who reported alcohol consumption (26.5%, 95% CI 18.0% to 35.9%, 3 studies).

### Comparative prevalence by place of residence

Six studies from India included non-slum populations alongside data from the slum population, and reported prevalence of hypertension by place of residence.[36 38 46 48 49 51] As shown in figure 5, the pooled prevalence of hypertension was highest among those residing in non-slum urban areas (33.5%, 95% CI 26.0% to 42.0%, 6 studies), followed by urban slum residents (28.8%, 95% CI 23.7% to 34.4%, 6 studies) and was lowest among rural residents (24.4%, 95% CI 18.4% to 31.5%, 5 studies). Slum residents were 35% more likely to be

hypertensive than those living in rural areas (OR=1.35, 95% CI 1.29 to 1.42) and 30% less likely to be hypertensive than those living in other urban areas (OR=0.70, 95% CI 0.51 to 0.96).

Four studies from India (n=3) and Bangladesh reported prevalence of type 2 diabetes by place of residence.[46 51 59 71] As shown in figure 6, the pooled prevalence of type 2 diabetes was highest among those residing in non-slum urban areas (13.06%, 95% CI 6.53% to 24.43%, 4 studies; 2813 participants), followed by urban slum residents (7.88%, 95% CI 3.32% to 17.55%; 4 studies; 1811 participants) and was lowest among rural residents (1.64%; 95% CI 0.06% to 32.21%; 3 studies; 405 participants). Prevalence of type 2 diabetes tended to be higher among urban slum residents than those living in rural areas (OR=3.78, 95% CI 0.75 to 18.93). Urban slum residents were 46% less likely to be diabetic than those from other urban areas (OR=0.54, 95% CI 0.44 to 0.66).

### Treatment cascade

Among those diagnosed with hypertension, only one-third were aware of their hypertensive status (33.6%, 95% CI 19.1% to 50.0%, 12 studies) (table 1). Among those aware of their high BP, half of them were on antihypertensive medications (51.9%, 95% CI 35.2% to 68.3%, 9 studies). Among those on treatment, only one-quarter had good BP control (25.2%, 95% CI 18.4% to 34.3%, 8 studies). Among those diagnosed with type 2 diabetes, 57.4% were aware of their type 2 diabetes status (95% CI 18.2% to 91.8%, 2 studies).

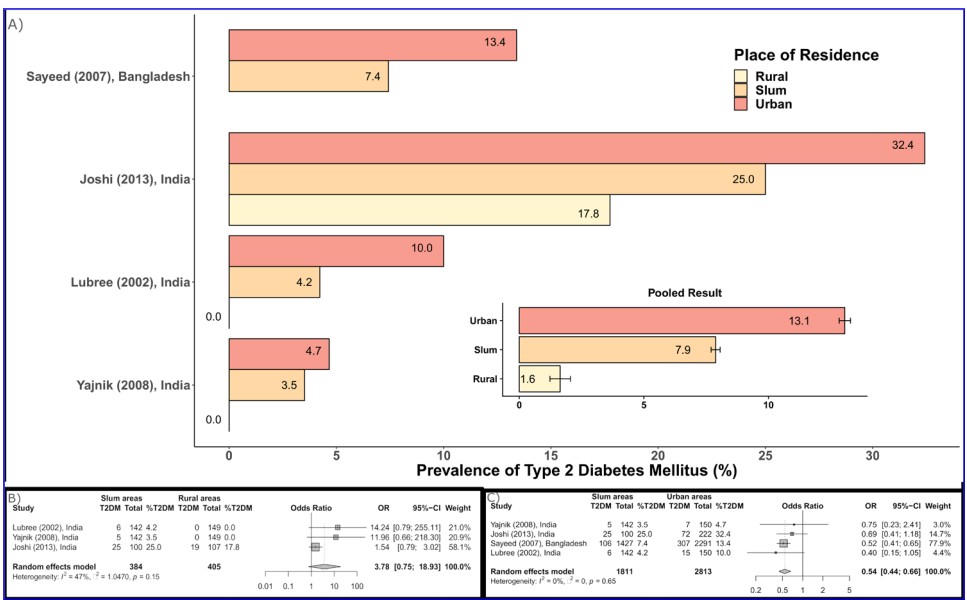

**Figure 6** Type 2 diabetes mellitus (T2DM) prevalence estimates by place of residence: urban versus rural versus slum. (A) Data from each studies, (B) Pooled estimates by place of residence, (C) Comparative pooled estimates.

## DISCUSSION
### Main findings

This systematic review and meta-analysis summarises available evidence on the global prevalence of hypertension and type 2 diabetes among slum residents. There were several key findings: first, the burden of hypertension and type 2 diabetes among slum dwellers is high and may be rising globally, with wide variation between countries and regions and, to some degree, also within countries. Using data from within-study comparator populations when presented, the pooled prevalence of hypertension and type 2 diabetes was highest among those residing in non-slum urban areas, followed by slum residents, and was lowest among rural residents. This finding corroborates those of previous reviews that observed higher prevalence of hypertension among urban residents than those living in rural areas.[81 82] This high prevalence may be due to rapid urbanisation, lifestyle changes, dietary changes and increased life expectancy,[83 84] or a combination of these factors.[85 86] In addition, the observed difference could be due to other factors including but not limited to lack of access to testing and care of NCD risk factors in rural areas and urban areas.

The observed gradient in burden of hypertension and type 2 diabetes among rural, slum and urban residents is consistent with the effects of urbanisation and wealth, as residents experience an economic transition when moving from one area to the next.[87–92] LMICs are now undergoing epidemiological transition, the change from a burden of infectious diseases to chronic diseases.[93] In addition, it could be due to increase in awareness in (non-slum) urban areas and recent availability of testing in some places. Recent systematic reviews of dietary risk behaviour in sub-Saharan Africa have found that urban populations tended to consume more salt than rural populations[94] and consume fewer portions of vegetables.[12] The rapid

pace of urbanisation and economic growth is accelerating the rate of this epidemiological transition; as such LMICs are at great risk of an explosive growth in the burden of NCDs, including hypertension and type 2 diabetes.[87 88]

We found evidence of significant unmet need for hypertension care among urban slum residents. A significant proportion of the urban slum residents were unscreened, undiagnosed, untreated or uncontrolled. This huge unmet need has been documented in previous studies from low/middle-income settings.[95–101] We also found that control of hypertension among slum residents was poor, such that only one in four slum residents on treatment had their BP controlled. The poor control of BP noted in our study, despite the fact the one-half of those who were unaware of high BP being on antihypertensive medications, needs further exploration. One possible explanation is availability and affordability of the medications and there could be minimal additional contact with a health professional.[15] It has been documented that the control of BP was related to the frequency of follow-up visits.[96] Another possible explanation could be low adherence to prescribed medications, as they may not be able to afford the medications.

As expected, we found that the burden of hypertension increased with the particpants' age, which may be attributed to age-related structural changes in blood vessels which potentially cause narrowing of the vascular lumen, and consequently increasing BP, as have been reported in previous studies.[102 103] The association between combined overweight/obesity and hypertension shown in our results exemplifies the role of excess body weight in hypertension prevalence, which has been long recognised and consistent across numerous observational and trial data.[104–106] We found evidence of significantly high prevalence of hypertension among smokers compared with non-smokers. Direct relation of chronic

tobacco consumption to hypertension however is not yet well established,[107 108] although tobacco consumption has been shown to cause an acute elevation of BP.[109]

## Study limitations and strengths

To the best of our knowledge, this paper is the first systematic review that summarises data about prevalence of hypertension and type 2 diabetes among slum residents. Strengths of this study include the use of a predefined and published protocol, a comprehensive search strategy and involvement of two independent reviewers in the review process. Nevertheless, the findings of this study should be interpreted with caution. Prevalence estimates from different regions and published over the course of 11 years were pooled in this meta-analysis, and as expected, high heterogeneity between studies was found in the meta-analyses. Nonetheless, as affirmed by previous evidence, meta-analyses are the preferred options to narrative syntheses for interpreting the results in a review, even in spite of the presence of a considerable amount of heterogeneity.[110] Heterogeneity appeared to be the norm rather than exception in published meta-analyses of observational studies.[111]

In conclusion, the burden of hypertension and type 2 diabetes varied widely between countries and regions and, to some degree, also within countries. In addition, many individuals with hypertension are not aware of their condition, not on treatment and control of hypertension is poor. The burden of hypertension and type 2 diabetes was higher among urban residents than their counterparts living in urban slums and rural areas. There is a need for public health strategies to improve the awareness, control and overall management of hypertension and type 2 diabetes in urban areas.

**Contributors** OAU, AA, OO and RJL conceived the study. OAU, AA and OO collected and analysed initial data. OAU, AA, OO, JS, PG and RJL participated in and contributed to refining the data analysis. OAU wrote the first manuscript. OAU, AA, OO, JS, PG and RJL contributed to further analysis, interpreting and shaping of the argument of the manuscript and participated in writing the final draft. OAU is the guarantor of this study.

**Funding** This research was funded by the National Institute for Health Research (NIHR) Global Health Research Unit on Improving Health in Slums using UK aid from the UK Government to support global health research (award ID: 16/136/87).

**Disclaimer** The views expressed in this publication are those of the author(s) and not necessarily those of the NIHR or the UK Department of Health and Social Care.

**Competing interests** None declared.

**Patient consent for publication** Not required.

**Ethics approval** This study does not involve human participants.

**Provenance and peer review** Not commissioned; externally peer reviewed.

**Data availability statement** All data relevant to the study are included in the article or uploaded as supplemental information.

for any error and/or omissions arising from translation and adaptation or otherwise.

**ORCID iDs**
Olalekan Abdulrahman Uthman http://orcid.org/0000-0002-8567-3081
Abimbola Ayorinde http://orcid.org/0000-0002-4915-5092

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
