## [Reviewer comments · BMJ Open]

ARTICLE DETAILS

TITLE (PROVISIONAL)	Global Prevalence and Trends in Hypertension and Type 2 Diabetes Mellitus among Slum Residents: A Systematic Review and Meta-analysis
AUTHORS	Uthman, Olalekan; Ayorinde, Abimbola; Oyeboode, Oyinlola; Sartori, Jo; Gill, Paramjit; Lilford, RJ

VERSION 1 – REVIEW

REVIEWER	Saif-Ur-Rahman, KM ICDDR, Health Systems and Population Studies Division
REVIEW RETURNED	23-Jun-2021

GENERAL COMMENTS	I appreciate the effort of the review authors for the interesting article. The article is methodologically sound, but there are some disparities. There is scope for improvement. Please find my comments here: 1. Page 7 (Line 124 -126): “Types of studies: We included all studies (cross-sectional studies, retrospective or prospective cohort studies) that reported prevalence of hypertension among slum residents as a primary or secondary outcome” What about type 2 DM? The title and results mentioned type 2 DM. Why is that missing from the inclusion criteria?2. Page 8 (Line 153-158): Authors have mentioned the title and abstract screening only? What about the full-text screening? Full-text screening is an essential part.3. Authors mentioned that they used the ROBANS tool to assess the risk of bias. Surprisingly, they have not used the domains mentioned in ROBANS. Please find the following article to use ROBANS appropriately: “Testing a tool for assessing the risk of bias for nonrandomized studies showed moderate reliability and promising validity (https://pubmed.ncbi.nlm.nih.gov/23337781/)” As per ROBANS, the domains for assessing the risk of bias are selection, performance, detection, attrition, and reporting. The risk of bias domains mentioned in Box 1 does not match that of ROBANS.4. Authors have missed one article which described the prevalence of both hypertension and type 2 DM among slum dwellers in Bangladesh. Please find the article here: “Prevalence of non-communicable disease risk factors among poor shantytown residents in Dhaka, Bangladesh: a community-based
---

	cross-sectional survey (https://www.ncbi.nlm.nih.gov/pmc/articles/PMC5695399/). Please add this article to the analysis and assess the risk of bias. 5. Please provide PRISMA 2020 checklist.
REVIEWER	Rajbhandari, Bibek Nepal Police Hospital, Emergency
REVIEW RETURNED	11-Aug-2021
GENERAL COMMENTS	Great manuscript, no place for negative comment. Manuscript as per PRISMA reporting guideline.

VERSION 1 – AUTHOR RESPONSE

Reviewer: 1

Dr. KM Saif-Ur-Rahman, Nagoya University Graduate School of Medicine Faculty of Medicine, ICDDRB Comments to the Author:

I appreciate the effort of the review authors for the interesting article. The article is methodologically sound, but there are some disparities. There is scope for improvement. Please find my comments here:

Authors' Response: Thanks for the pertinent comments.

1. Page 7 (Line 124 -126): "Types of studies: We included all studies (cross-sectional studies, retrospective or prospective cohort studies) that reported prevalence of hypertension among slum residents as a primary or secondary outcome"

What about type 2 DM? The title and results mentioned type 2 DM. Why is that missing from the inclusion criteria?

Authors' Response: Thanks, we have now clarified this "We included all studies (cross-sectional studies, retrospective or prospective cohort studies) that reported prevalence of hypertension and type 2 diabetes mellitus among slum residents as a primary or secondary outcome."

We also include this in the types of outcomes:

"Type 2 diabetes was defined based on measured fasting plasma glucose, or oral glucose tolerance test. Type 2 diabetes was diagnosed if the fasting blood glucose was ≥ 126 mg/dL (≥ 7.0 mmol/L) after an overnight fast for at least 8 hours, or random capillary blood glucose of ≥ 11.1 mmol/L or if the participant was taking treatment for type 2 diabetes."

2. Page 8 (Line 153-158): Authors have mentioned the title and abstract screening only? What about the full-text screening? Full-text screening is an essential part.

Authors' Response: Thanks, we have now clarified this:

"In pairs, three reviewers (OAU, AAA, OO) independently then independently evaluated the full-text articles of all identified citations to establish relevance of the article according to the pre-specified criteria. In cases of discrepancies, agreement was reached by discussion with a third reviewer."

3. Authors mentioned that they used the ROBANS tool to assess the risk of bias. Surprisingly, they have not used the domains mentioned in ROBANS. Please find the following article to use ROBANS appropriately:

“Testing a tool for assessing the risk of bias for nonrandomized studies showed moderate reliability and promising validity (<https://pubmed.ncbi.nlm.nih.gov/23337781/>)”

As per ROBANS, the domains for assessing the risk of bias are selection, performance, detection, attrition, and reporting. The risk of bias domains mentioned in Box 1 does not match that of ROBANS. Authors’ Response: Thanks for pointing this out, this has been corrected.

“The risk of bias of included studies will be assessed by using the Strengthening the Reporting of Observational Studies in Epidemiology (STROBE)”

Most of the RoBANS items is not relevant for cross-sectional studies.

4. Authors have missed one article which described the prevalence of both hypertension and type 2 DM among slum dwellers in Bangladesh. Please find the article here:

“Prevalence of non-communicable disease risk factors among poor shantytown residents in Dhaka, Bangladesh: a community-based cross-sectional survey (<https://www.ncbi.nlm.nih.gov/pmc/articles/PMC5695399/>)”. Please add this article to the analysis and assess the risk of bias.

Authors’ Response: Thanks, we have now included the study and re-analysed all the pooled prevalence estimates.

5. Please provide PRISMA 2020 checklist.

Authors’ Response: This was included as part of the annexes; we have uploaded this as a separate document.

Reviewer: 2

Dr. Bibek Rajbhandari, Nepal Police Hospital Comments to the Author:

Great manuscript, no place for negative comment.

Manuscript as per PRISMA reporting guideline.

Authors’ Response: Thanks for the comments.

VERSION 2 – REVIEW

REVIEWER	Saif-Ur-Rahman, KM ICDDR, Health Systems and Population Studies Division
REVIEW RETURNED	26-Nov-2021

GENERAL COMMENTS	I appreciate the effort of the review authors for the interesting article. However, in the revision, the methodological issues are not addressed. Please find my comments here: 1. Authors have mentioned that “In pairs, three reviewers (OAU, AAA, OO) independently then independently evaluated the full-text articles of all identified citations to establish the relevance of the article according to the pre-specified criteria. In cases of discrepancies, the agreement was reached by discussion with a third reviewer.” This is not clear. Usually, the screening is done by two independent review authors. I don’t understand how three authors
---

	did that independently. Even if they do it in pairs, there should be four review authors. This poses a question to the claim of independent screening. 2. Previously I mentioned that the authors have not used the domains mentioned in ROBANS. Please find the following article to use ROBANS appropriately: “Testing a tool for assessing the risk of bias for nonrandomized studies showed moderate reliability and promising validity (https://pubmed.ncbi.nlm.nih.gov/23337781/)” As per ROBANS, the domains for assessing the risk of bias are selection, performance, detection, attrition, and reporting. The risk of bias domains mentioned in Box 1 does not match that of ROBANS. The authors have responded that this has been corrected. “The risk of bias of included studies will be assessed by using the Strengthening the Reporting of Observational Studies in Epidemiology (STROBE)” Most of the RoBANS items are not relevant for cross-sectional studies. Authors have mentioned that they corrected as per STROBE, but kept the tables as before (eTable 3: Risk of bias of included studies). This is not acceptable. In addition, this is to mention that STROBE is not a tool for assessing the risk of bias. Authors may use RoBANS for assessing the risk of bias. Alternatively, they might use the JBI tool for the quality assessment of prevalence studies. My suggestion is to use RoBANS incorporating the appropriate domains mentioned in the manuscript that I referred to. I do not agree with the statement of the authors that “most of the RoBANS items are not relevant for cross-sectional studies”. 3. Box 1 mentioned in the manuscript is not relevant and can be added as supplementary material.
--	---

VERSION 2 – AUTHOR RESPONSE

Reviewer: 1

Dr. KM Saif-Ur-Rahman, Nagoya University Graduate School of Medicine Faculty of Medicine, ICDDRB

1. Authors have mentioned that “In pairs, three reviewers (OAU, AAA, OO) independently then independently evaluated the full-text articles of all identified citations to establish the relevance of the article according to the pre-specified criteria. In cases of discrepancies, the agreement was reached by discussion with a third reviewer.”

This is not clear. Usually, the screening is done by two independent review authors. I don’t understand how three authors did that independently. Even if they do it in pairs, there should be four review authors. This poses a question to the claim of independent screening.

Authors' reply: Contrary to the reviewer's belief and assertion that it is not possible for three reviewers to screen the citations in pairs, it is absolutely possible. We had to think outside the box and use innovative combinations.

We would like to explain how it is possible for screening to be done in pairs by three reviewers. For example, we screened 1306 citations in pairs by three of us as follow: The total number of citations were divided into two, the first reviewer screen all citations and the second and third reviewers screened half-half of the citations.

Therefore, it was done in pairs by three reviewers, ...QED. It was very easy and straight forward logic as shown in the table below:

Citation number	Reviewer 1	Reviewer 2
-----------------	------------	------------

1-653	OAU	AAA
-------	-----	-----

654-1306	OAU	OO
----------	-----	----

Discussion with the third reviewer.

Citation number	Reviewer 1	Reviewer 2	Reviewer 3
-----------------	------------	------------	------------

1-653	OAU	AAA	OO
-------	-----	-----	----

654-1306	OAU	OO	AAA
----------	-----	----	-----

However, in order to avoid any further confusion, we have now modified the methods section as "Two reviewers (OAU, AAA) independently evaluated the eligibility and methodological quality of the studies obtained from the literature searches. All articles yielded by the database search were initially screened by their titles and abstracts to obtain studies that met inclusion criteria. In cases of discrepancies, agreement was reached by discussion with a third reviewer. Two reviewers (OAU, AAA) independently then independently evaluated the full-text articles of all identified citations to establish relevance of the article according to the pre-specified criteria. In cases of discrepancies, agreement was reached by discussion with a third reviewer."

2. Previously I mentioned that the authors have not used the domains mentioned in ROBANS. Please find the following article to use ROBANS appropriately:

“Testing a tool for assessing the risk of bias for nonrandomized studies showed moderate reliability and promising validity (<https://pubmed.ncbi.nlm.nih.gov/23337781/>)”

As per ROBANS, the domains for assessing the risk of bias are selection, performance, detection, attrition, and reporting. The risk of bias domains mentioned in Box 1 does not match that of ROBANS.

The authors have responded that this has been corrected.

“The risk of bias of included studies will be assessed by using the Strengthening the Reporting of Observational Studies in Epidemiology (STROBE)”

Most of the RoBANS items are not relevant for cross-sectional studies.

Authors have mentioned that they corrected as per STROBE, but kept the tables as before (eTable 3: Risk of bias of included studies). This is not acceptable. In addition, this is to mention that STROBE is not a tool for assessing the risk of bias. Authors may use RoBANS for assessing the risk of bias. Alternatively, they might use the JBI tool for the quality assessment of prevalence studies. My suggestion is to use RoBANS incorporating the appropriate domains mentioned in the manuscript that I referred to. I do not agree with the statement of the authors that “most of the RoBANS items are not relevant for cross-sectional studies”.

Authors’ reply: As requested by the reviewer, we have now used ROBANS.

This has been modified in the methods, results and supplementary material.

“We used the Risk of Bias Assessment tool for Non-randomized Studies (RoBANS)²⁴ to assessed the risk of bias of included studies (see Box 1). The risk of bias in a study was graded as low, high or unclear on the basis of study features including the selection (selection of participants and confounding variables), performance (measurement of exposure), detection (blinding of outcome assessments), attrition (incomplete outcome data) and reporting (selective outcome reporting).”

Box 1 mentioned in the manuscript is not relevant and can be added as supplementary material.

Authors’ reply: Thanks, this has now been moved to the supplementary material

VERSION 3 – REVIEW

REVIEWER	Saif-Ur-Rahman, KM ICDDR, Health Systems and Population Studies Division
REVIEW RETURNED	09-Dec-2021
GENERAL COMMENTS	Thanks for addressing my comments. I appreciate the effort of the authors. There are some typological errors, please correct those during the proofreading.